# MicroRNAs in Animal Models of HCC

**DOI:** 10.3390/cancers11121906

**Published:** 2019-12-01

**Authors:** Francesca Fornari, Laura Gramantieri, Elisa Callegari, Ram C. Shankaraiah, Fabio Piscaglia, Massimo Negrini, Catia Giovannini

**Affiliations:** 1Center for Applied Biomedical Research, St. Orsola-Malpighi University Hospital, 40138 Bologna, Italy; fabio.piscaglia@unibo.it (F.P.); catia.giovannini4@unibo.it (C.G.); 2Department of Medical and Surgical Sciences, University of Bologna, 40138 Bologna, Italy; 3Department of Morphology, Surgery and Experimental Medicine, University of Ferrara, 44121 Bologna, Italy; elisa.callegari@unife.it (E.C.); shnrch@unife.it (R.C.S.);

**Keywords:** HCC, microRNA, animal models

## Abstract

Hepatocellular carcinoma (HCC) is the second leading cause of cancer-related mortality. Molecular heterogeneity and absence of biomarkers for patient allocation to the best therapeutic option contribute to poor prognosis of advanced stages. Aberrant microRNA (miRNA) expression is associated with HCC development and progression and influences drug resistance. Therefore, miRNAs have been assayed as putative biomarkers and therapeutic targets. miRNA-based therapeutic approaches demonstrated safety profiles and antitumor efficacy in HCC animal models; nevertheless, caution should be used when transferring preclinical findings to the clinics, due to possible molecular inconsistency between animal models and the heterogeneous pattern of the human disease. In this context, models with defined genetic and molecular backgrounds might help to identify novel therapeutic options for specific HCC subgroups. In this review, we describe rodent models of HCC, emphasizing their representativeness with the human pathology and their usefulness as preclinical tools for assessing miRNA-based therapeutic strategies.

## 1. MicroRNAs in HCC

MicroRNA (miRNA) profiles are highly informative for tumor classification and prognosis [1,2]. In line, the aberrant expression of miRNAs represents a hallmark of human hepatocellular carcinoma (HCC); miRNAs signatures associated with increased HCC risk, neoplastic development, advanced stages, and vascular invasion, were identified in human HCC, and some of them were confirmed in rodent models and tested as therapeutic targets [3,4,5,6,7,8] (Figure 1). From a mechanistic perspective, functional studies have demonstrated the active involvement of specific miRNAs in the regulation of key pathways driving hepatocarcinogenesis, through the direct targeting of crucial oncogenes or tumor-suppressor genes [3,9,10,11,12,13,14]. Besides a restricted panel of informative miRNAs shared by different cohorts, a remarkable inconsistency can be observed across independent studies. This inconsistency may be ascribed to the heterogeneous prevalence of etiologies, environmental and genetic factors, and technical and analytical reasons. The molecular heterogeneity of HCC indeed still represents a critical factor. A lack of biomarkers classifying HCC subgroups still limits our ability to allocate patients to the best therapeutic option. In this perspective, identifying common molecular traits shared by specific HCC subgroups and animal models might help to deepen molecular knowledge, favoring the discovery of novel diagnostic and therapeutic approaches in molecularly defined contexts. Due to the putative roles of miRNAs as molecular biomarkers (both tissue and circulating) as well as therapeutic targets or therapeutic molecules, this class of RNAs deserves attention as a useful tool at multiple levels.

## 2. Animal Models and HCC Research

More than 80% of human HCCs arise in a field of chronic liver disease resulting from viral hepatitis, alcohol abuse, metabolic syndrome, exposure to carcinogenic agents, and genetic diseases such as Wilson’s disease or hemochromatosis. Different etiologies, together with different ethnicities, contribute to the high HCC heterogeneity. Since molecular inter-tumor and intra-tumor heterogeneity is a matter of fact in HCC [28], identifying subclasses with predictable prognosis and response to treatments is one of the major challenges in the clinical management of this disease. The development of animal models that recapitulate in vivo the multistep natural history of HCCs might help in understanding the causes, as well as provide the rationale for testing treatments. The “ideal” animal model should mimic human HCC natural history. However, no model is “ideal” for all HCC research purposes, and spontaneous models of tumorigenesis are rare. Because of the genetic similarities between rodents and humans, the breeding capacity, and the short lifespan, rodents are widely used for cancer research. A wide range of models is currently available to study HCC development and therapeutic treatments, including chemically induced, transgenic, or knockout mice; xenograft mice; and dietary-induced models, such as the choline-deficient diet. In addition, combinations of different models have also been used. Beside mice and rats, zebrafish (*Danio rerio*) also deserve attention due to a series of characteristics that will be briefly described. Since there is no single dominant molecular driver underlying all HCCs, specific models should be chosen according to their representativity of specific subtypes of HCCs. Genomic and transcriptomic studies classify HCC based on molecular pathway activation. Comparison of gene expression between each animal model and human HCC subgroups may help to choose the ideal model for specific aims. Here, we review some of the most representative HCC animal models selected on the basis of the following:-The extent of representativeness and the relevance of miRNA deregulation in human HCC;-The translational relevance of miRNA-based therapeutic intervention in rodent models.

## 3. miRNAs Knockout and Transgenic Mice

The development of genetically engineered mouse (GEM) models greatly contributed to the advancement of knowledge of pathogenic mechanisms sustaining human diseases, including cancer. The great advantage of GEM is that genetic manipulations can be introduced systematically, in a specific organ, but also at defined time points during development, growth, and aging. An exhaustive description of conditional and inducible expression systems has been reviewed by He and coworkers [29]. GEM models also provide an optimal background for preclinical studies testing molecularly based treatments. miRNA-based knockout (KO) or transgenic (TG) models will be hereafter briefly described.

## 4. KO Mouse Models

### 4.1. Dicer1 KO Mouse Model

To investigate the consequences of miRNA disruption in liver physiology, *Dicer1* conditional knockout mice were established and showed initial preservation of liver function, followed by the development of progressive liver dysfunctions leading to prominent steatosis, lipid, and glucose metabolism impairment, increased hepatocyte proliferation, and, finally, onset of HCC [30,31]. These studies reported the downregulation of the hepatospecific miR-122 that accounts for 70% of all expressed miRNAs in normal liver, which role in cholesterol biosynthesis and lipid metabolism is well established [32,33]. Interestingly, despite liver repopulation by Dicer positive cells at 12-month after birth, two-third of *Alb-Cre*; *Dicer (fl/fl)* animals displayed liver tumors exhibiting decreased levels of *Dicer1* and lack of miR-122 expression with respect to non-neoplastic surrounding tissue, demonstrating the loss of dicer as a driver event in hepatocarcinogenesis [30]. In line, the downregulation of miRNA machinery components (*Dicer1* and *p68*) is a frequent event in HCC and it is associated with decreased tumor-free survival [34].

### 4.2. miR-122 KO Mouse Model

Several lines of evidence highlight the importance of miR-122 in liver biology and disease, as suggested by its downregulation in most of HCC tissues and its peculiar role in noncanonical regulation of hepatitis C virus (HCV) life cycle, as well as by the modulation of genes involved in the maintenance of the adult liver phenotype [3,32,35]. In this scenario, germline and liver-specific conditional *miR-122*-KO mice were developed in order to investigate the consequences of its genetic loss [36,37]. Both *miR*-*122*-KO and liver-specific KO (LKO) mice developed microsteatosis and liver inflammation, characterized by increased lipid droplets and infiltrating cells secreting inflammatory cytokines (IL-6 and TNF-α), that progressed to steatohepatitis, fibrosis and spontaneous HCC. Notably, an enrichment of transcripts bearing a miR-122 complementary binding site in their 3’UTR was detected in LKO mice and ingenuity pathway analysis (IPA) revealed the altered expression of genes belonging to triglyceride biosynthesis, as well as oncogenes, commonly associated with malignant transformation, such as insulin-like growth factor 2 (Igf2), beta catenin (Ctnnb1), Kirsten rat sarcoma viral oncogene homolog (Ras), epithelial cell adhesion molecule (Epcam), and avian myelocytomatosis virus oncogene cellular homolog (c-Myc). In addition, a signature of genes specifically deregulated in *miR-122*-KO and LKO mice was able to stratify human tumor samples into high or low miR-122 expression groups, suggesting these preclinical models as representative of the human pathology, mimicking key features of cancer progression in a background of fatty liver disease [36]. Similarly, the reactivation of embryo-specific genes and imprinted miRNA clusters was reported following adeno-associated virus (AAV)-mediated miR-122 transient disruption in a mouse model. In particular, several miRNAs located in *Igf2* and *Dlk1-Dio3* imprinted loci were markedly activated following miR-122 loss in liver tissue from both transient and stable *miR-122*-KO mice, suggesting initiation of precancerous transcriptional modifications in mice lacking miR-122 expression [38,39]. These studies emphasize the intriguing idea of complex regulatory networks and inter-relationships between miRNAs, transcription factors, epigenetic modulators and miRNA machinery complexes whose stringent cooperation defines the precancerous transcriptional phenotype observed in miR-122^−/−^ mice that is, strikingly, shared by human HCCs.

Notably, in vivo gene targeting of *Dlk1-Dio3* locus by AAV vectors causing its overexpression led to HCC development in 100% of mice and, in line, overexpression of this miRNA cluster associated with an aggressive stem-cell-like phenotype in HCC [40]. Others and our group reported the upregulation of miR-494, a member of the *Dlk1-Dio3* miRNA cluster, in 25–30% of HCCs with stemness features and demonstrated its involvement in tumor progression and sorafenib resistance through the direct targeting of mutated in colorectal cancer (*MCC*) and the phosphatase and tensin homolog (*PTEN*) tumor suppressor gene [15,16]. Regarding *IGF2/483* oncogenic locus, an increase of miR-483-3p was found in 30% of human HCCs and Bcl-2 binding component 3 (*BBC3/PUMA*), a Bcl-2 family member with pro-apoptotic functions, was identified among its target genes, suggesting miR-122 KO models as representative of a subgroup of HCCs with stem-cell-like characteristics and epigenetic deregulation [41].

### 4.3. miR-148a KO Mouse Model and Epigenetic Changes

Knockout mice for miR-148a, a liver abundant miRNA, were established to assess its intrinsic role in liver physiology and to characterize its contribution to hepatocarcinogenesis. Indeed, miR-148a-3p/5p decreased expression was observed in tumor versus nontumor tissue of HCC patients, and it was associated with poor differentiation and reduced overall survival. miR-148a-KO mice showed an increase of steatosis and serum total cholesterol levels, as well as an upregulation of genes involved in lipogenesis and fatty acid uptake, when fed with both a regular-chow diet (RCD) and high-fat diet (HFD). As expected, several genes involved in lipid and cholesterol biosynthesis were identified among miR-148a-3p/5p targets displaying increased levels in KO animals, independently of the diet. Moreover, tumor induction by a single diethylnitrosamine (DEN) injection led to an increased number and size of tumor nodules in KO mice, whereas restoring pri-miR-148a levels reduced tumor count and volume [17]. Interestingly, miR-148a progressive upregulation was detected in a mouse fetal hepatoblast model when subjected to liver differentiation, which corresponded to an opposite decrease of its target gene DNA methyltransferase 1 (DNMT1). Functional analysis demonstrated the involvement of miR-148a/DNMT1 axis in triggering expression of adult hepatic markers, albumin (Alb), glucose-6-phosphatase catalytic subunit (G6PC), and tyrosine aminotransferase (Tat), and proved a negative feedback loop between miR-148a and *DNMT1* itself. Notably, miR-148a decreased expression was detected in liver biopsies from HCC patients with respect to normal livers, but not surrounding tissues, suggesting its involvement in the progression of the underlying liver disease. Finally, gain-and-loss of function studies demonstrated its role in preventing invasive capabilities of HCC cells through mesenchymal epithelial transition factor (c-Met) indirect targeting and reported the oncogene c-Myc among miR-148a transcriptional inhibitors contributing to its downregulation during hepatocarcinogenesis [42].

### 4.4. miR-223 KO Mouse and NAFLD

A recent study described the protective activity of the neutrophil-associated miRNA, miR-223, in nonalcoholic steatohepatitis (NASH) and HCC, by the direct targeting of inflammatory and oncogenic genes upregulated in these pathologic conditions. HFD-fed C57BL/6J mice developed steatosis but were resistant toward NASH progression; strikingly, higher tissue levels of the anti-inflammatory miR-223 were found in hepatocytes from HFD-fed mice, as well as in liver specimens from NASH patients, with respect to control-diet-fed animals and healthy liver samples, respectively. miR-223KO mice developed a full spectrum of nonalcoholic fatty liver disease (NAFLD) and more severe NASH phenotypes, as corroborated by higher levels of serum alanine aminotransferase (ALT), greater liver infiltration and fibrosis, and increased mRNA levels of pro-inflammatory cytokines. In addition, IPA of microarray data revealed the dysregulation of genes contributing to carcinogenesis and inflammatory response in KO with respect to wild type (WT) mice after three months of HFD. In line, half of miR-223KO animals developed HCC after long-term HFD feeding, showing an increased susceptibility to disease progression. Since miR-223 positively correlated with several chemokines (C-X-C motif chemokine 10, *CXCL10*) and cytokines (interleukin-6, *IL-6*) and with cancer-related genes (glypican 3, *GPC3*) in NASH patients, the authors hypothesized a protective role for miR-223 against malignant transformation and disease progression, contributing to the explanation as to why HFD feeding alone is not sufficient for NAFLD progression in mouse models [43]. These interesting findings depicting the anti-inflammatory role of miR-223 in NASH patients agree with data from other groups, as wee as our own, showing a downregulation of miR-223 in HCC with respect to surrounding nontumor tissues from surgically resected HCCs, corroborating the idea that high miR-223 levels restrain tumor development [3,44].

## 5. Transgenic Mouse Models

### 5.1. miR-221 TG Mouse

miR-221 is upregulated in 70–80% of HCCs and in a wide range of different malignancies. The recognition of miR-221 related targets, such as the cyclin-dependent kinase inhibitors CDKN1B/p27 and CDKN1C/p57, PTEN, or the pro-apoptotic protein Bcl2 modifying factor (BMF), explains several mechanisms through which its upregulation contributes to carcinogenesis [9,10,14]. We developed a liver-specific miR-221 transgenic mouse model (TG221) which spontaneously develops HCC in 50% of male mice aged 9–12 months. DEN administration accelerates the appearance of liver cancer that becomes visible at 3–4 months of age in all male mice. Interestingly, in this TG model, the degree of miR-221 upregulation quantitatively matches the extent of miR-221 deregulation in human HCC, with a two-to three-fold change when comparing HCC with surrounding tissue. Specifically, following intraperitoneal DEN injection TG221 animals displayed a higher liver weight associated with increased number and volume of tumor nodules [18]. Lower expression of miR-221 validated targets, p27, p57, and Bmf in TG tumors and liver samples with respect to WT animals confirmed this model as representative of molecular mechanisms previously identified in HCC cell lines and human tumors [10]. This model was employed to test in vivo the antitumor activity of chemically modified anti-miR oligonucleotides (AMOs). Briefly, the experimental protocol consisted in one intraperitoneal (IP) injection of DEN in 10-day newborn TG mice and, two months later, three intravenous (IV) injections of anti-miR-221 every 15 days. A reduction of size and number of tumors was detected in the group receiving miR-221 AMOs with respect to control-treated mice, showing a persistent decrease of miR-221 levels and demonstrating the specificity and efficacy of this therapeutic strategy. Therefore, the TG221 model confirmed miR-221 as a driver gene for HCC development and progression and resulted as an effective tool to test miRNA-mimic or anti-miRNA-based approaches increasing the knowledge regarding their safety, specificity, and efficacy for the treatment of HCC. In this regard, the expression of several HCC-specific miRNAs, miR-199a-3p, miR-122, and miR-21 in TG221 mice reflected those observed in primary human HCCs; therefore, this model was employed to assay the anticancer activity of miRNA mimic oligonucleotides too [18]. Besides anti-miR-221, miR-199a/b-3p mimics were tested to treat HCCs developed in miR-221 TG mouse. miR-199a-3p represents the third most abundantly expressed miRNA in the liver, and it is downregulated in virtually all HCCs with a fold change greater than 10-fold in many cases, representing an ideal candidate for in vivo replacement therapy [3,45,46,47]. It controls several cancer-associated genes, such as mammalian target of rapamycin (*mTOR*), hepatocyte growth factor receptor *c-Met*, the kinase p21-activated kinase 4 (*PAK4*), and the Notch regulator Yes-associated protein 1 (*YAP1*). Since an miR-221 TG mouse also displays miR-199a-3p constitutive downregulation, miR-199a-3p mimics were adopted by using two experimental protocols: DEN priming in 10-day newborn mice followed by three IP injections per week for three weeks at three or five months of age. Both protocols showed a decreased number and size of nodules in miR-199a-3p mimics treated group with an anticancer effect similar to that observed in sorafenib-treated mice [19]. A decrease of mTOR and PAK4 protein expression was observed in HCC specimens from miRNA mimics treated animals, confirming similar mechanisms of action observed in other preclinical models, as well as in human HCCs [3,46]. An interesting finding is that the livers of miR-221 TG mice are characterized by a relevant steatohepatitis, which resembles the human NASH that is becoming a more and more prevalent risk factor for HCC [48].

### 5.2. miR-221 TG Mouse Developing HCC on Liver Cirrhosis

One limitation of most HCC murine models is the absence of cirrhosis, which, instead, accompanies the multistep natural history of human HCC in about 90% of cases, with the well-known role of associated microenvironmental changes. Genetic manipulations may modify the expression of key molecules driving liver carcinogenesis allowing studying specific mechanisms, which, however, take place on a healthy liver. The preneoplastic stage of cirrhosis represents a background to be kept in mind when trying to match animal models with human disease. Indeed, cirrhosis is a distinct disease to face with and its own contribution through microenvironmental changes should be considered when assaying novel biomarkers or dissecting molecular mechanisms. This is true not only for molecular purposes, but also for functional reasons when testing the therapeutic/toxic effect of drugs. Notwithstanding this, the animal models of HCC arisen on a background of overt cirrhosis are very few. Among these, the HCC mouse model developed by Domenicali and co-authors provides the background for repeated cycles of liver damage/regeneration to reflect faithfully the events leading to the development of cirrhosis [49] This model recapitulates the progression from hepatocyte cell death, proliferation, fibrosis, towards cirrhosis with portal hypertension syndrome. Notably, only a very restricted fraction of these animals develops HCC. Briefly, a well-tolerated protocol able to induce cirrhosis in C57BL/6N mice consisted of administration of the carbon tetrachloride (CCl4) by short inhalation cycles, twice or three times weekly, administered through increasing exposure times, resulting in a safe and effective method to induce decompensated cirrhosis with ascites. Conversely, subcutaneous or intraperitoneal CCl4 administration displayed adverse events, preventing the development of an overt cirrhosis with portal hypertension syndrome. Thus, Callegari et al. applied this model in mice bearing a genetic factor that predisposed them to HCC, such as the miR-221 TG mouse [18]. Taking advantage of the protocol developed by Domenicali et al. [49], multiple short-term inhalation cycles of CCl4 were used to induce decompensated cirrhosis in TG221 transgenic mice, to test the possibility of inducing liver tumors in the context of cirrhosis. Unlike the very low incidence of HCC in control mice developing cirrhosis, when applied to TG221 mice at four weeks from the end of CCl4 treatment, ultrasound analysis revealed large liver nodules which were histopathologically confirmed as dysplastic nodules or HCC, whereas, in WT mice, most were macro-regenerative or hyperplastic nodules, without dysplastic features. All animals displayed overt cirrhosis, demonstrated by histopathological examination, by the presence of ascites, and by activation of fibrotic markers such α-smooth muscle actin (*α-SMA*), connective tissue growth factor (*CTGF*), and transforming growth factor beta 1 (*TGF-1*). Notably, in TG221 mice, necro-inflammatory changes were more intense and extended, and progression to dysplastic nodules or HCCs was confirmed based on morphology, immunohistochemistry (increased expression of tumor-associated genes such as alpha-fetoprotein ‘*AFP*’ and ‘*GPC3*’) and gene expression patterns. This model appears to recreate faithfully the natural history of HCC on a background of cirrhosis. It could also be considered to investigate microenvironmental changes associated with disease progression and to achieve functional implications relevant for drug metabolism and toxicity. In conclusion, the cirrhosis-associated TG221 model represents a robust preclinical tool for testing therapeutic or prophylactic strategies in a preclinical setting recapitulating the pathophysiology of human HCC [20].

### 5.3. miR-17-92 TG Mouse

miR-17-92, together with its paralogous miR-25-106b cluster, is one of the first polycistronic miRNA clusters whose overexpression was recognized in hematologic and solid tumors [50,51]. miR-17-92 is transcriptionally regulated by c-Myc proto-oncogene, and it is required to maintain a neoplastic state through the regulation of genes involved in chromatin remodeling and apoptosis [52,53]. In order to dissect the contribution of miR-17-92 cluster to hepatocarcinogenesis, Zhu and coworkers investigated its expression in HCC patients and established a transgenic mouse model overexpressing the six-miRNA family members in the liver. Quantitative polymerase chain reaction (QPCR) analysis and in situ hybridization (ISH) were performed in two HCC patient cohorts reporting miRNA (miR-17, miR-18, miR-19a, miR-19b, miR-20, and miR-92) overexpression in tumor tissues with respect to nontumor specimens; miRNA sequencing data from the Cancer Genome Atlas (TCGA) database further confirmed their overexpression in a larger cohort of patients. The miR-17-92 TG mouse model showed augmented propensity to liver cancer induction following DEN administration in comparison to WT mice, as evidenced by an increase of tumor nodule number and volume at 10-month. In addition, TG mice displayed a faster tumor progression, showing a higher percentage of HCC nodules [54].

### 5.4. HBx Transgenic Mouse and miR-224

An intriguing mechanism of miRNA degradation was proposed for miR-224 whose expression negatively correlated with autophagy in HBV-related HCCs, as demonstrated by the tissue array of autophagy-associated markers (autophagy related 5 ‘*Atg5*’, *Beclin1*, and *p62*) and ISH for miRNA detection in a cohort of 93 HCC patients. In particular, a negative correlation between miR-224 and autophagy-promoter gene *Atg5*, and a positive correlation between miRNA levels and p62 gene, whose accumulation typically characterizes autophagy impairment, were displayed in hepatitis B virus (HBV)-infected patients, but not in hepatitis C virus (HCV)-positive ones. To confirm this peculiar miR-224/autophagy relationship occurring in HBV-related HCCs and to clarify autophagy-mediated mechanisms of miRNA regulation, an Hepatitis B Virus protein X (HBx) transgenic mouse model was used [55]. The oncogenic properties of the HBx gene are indeed well-known and these TG animals, established in 1991, showed progressive histopathological liver changes progressing to carcinomas that were fatal to all the animals [56]; thus, they represent ideal models for defining molecular events involved in HBV-associated liver tumorigenesis. Low Atg5 expression accompanied by p62 accumulation and high miR-224 levels were detected in tumor specimens with respect to nontumor tissues of 17.5 months old mice. Notably, only mature miR-224 isoform but not precursor transcript was upregulated in HCC patients and HBx TG mice, suggesting post-transcriptional events participating to its regulation. Elegant in vitro experiments with autophagy inducers and inhibitors demonstrated miR-224 recruitment to the autophagosome and its degradation following fusion with lysosomes, independently from viral status of hepatoma cells, postulating that induction of autophagy or miR-224 antagonism are effective antitumor strategies in HCC. In addition, SMAD family member 4 (*Smad4*) and glycine-N-methyltransferase (*GNMT*) were described among miR-224 target genes, and their involvement in liver injury and tumor progression was proved by in vitro and in vivo models [55,57]. Again, miR-224 upregulation and inverse correlation with its target *GNMT* were described in HBV-related HCC patients only, confirming the virus-specificity for miR-224 aberrant expression [55,57]. Consistently, miR-224 upregulation characterized early stages of HBV-related hepatocarcinogenesis in different animal models, highlighting the necessity of proper animal models when virus-related miRNA-based therapeutic options are concerned [58].

Interestingly, Tang and coworkers compared gene-expression profiling between HBX-TG mice and chemically induced DEN-HCC mice. They showed that upregulated genes in tumor versus normal tissue were mainly involved in immune and acute-phase response and cholesterol and lipid biosynthesis in the DEN model, whereas upregulated genes belonged to positive regulation of gene expression, cell proliferation, migration and invasion, and immune response in the HBX model. On the contrary, both models shared common deregulated genes involved in metabolic pathways and redox processes. Moreover, early growth response 1 ‘*Egr1*’, activating transcription factor 3 ‘*Atf3*’ and Kruppel-like factor 4 ‘*Klf4*’ resulted as the main regulatory factors in HBX mice controlling cancer-associated signaling pathways, such as PI3K/AKT, MAPK, Ras, and p53, which role in hepatocarcinogenesis is well established [59]. Notably, several oncogenes and tumor-suppressor (TS) genes deregulated in human HCC were controlled by these transcription factors, highlighting the reliability of the HBX mouse model with respect to the human pathology.

## 6. Oncogene-Related TG Mice

*Myc* and *Ras* are canonical oncogenes whose increased expression, amplification, or pathway activation has been identified in most of human cancers. Regarding HCC, Ras pathway is often activated due to promoter hypermethylation of Ras association domain family member 1 (*RASSF1*) and novel Ras effector 1A (*NORE1A*) inhibitors [60]. Interestingly, *MYC* oncogene was recently associated with a gene regulatory network formed by lin-28 homolog B (*LIN28B*), *CTNNB1*, SWI/SNF Related, Matrix Associated, Actin Dependent Regulator of Chromatin, Subfamily A, Member 4 (*SMARCA4*), *Let-7b*, SRY-Box Transcription Factor 9 (*SOX9*), and *TP53* that correlated with increased proliferation, dedifferentiation and c-Met pathway activation more likely contributing to the transcriptional signature typical of the HCC “proliferation” subclass [61].

Oncogene-inducible liver-specific transgenic mouse model gave rise to liver tumors with a 100% penetrance within 12 weeks. Histological characterization revealed that *Myc*-driven tumors resemble poorly differentiated HCCs or human hepatoblastomas, *Ras*-driven tumors resemble human HCCs, and *Myc* + *Ras*-driven tumors mirrored an aggressive variant of HCC or fetal variant of hepatoblastoma. Molecular characterization showed the overexpression of chromosome 12qF1 miRNA megacluster, comprising 53 miRNAs, in *Myc*, *Ras,* and *Myc* + *Ras* TG mice and in a subgroup of human HCCs (human homolog located at chromosome locus 14q32). In particular, a 60-miRNA signature was shared by these three oncogene-dependent models and was representative of an HCC patients’ subgroup with high AFP expression, displaying 29 out of 60 members belonging to the *Dlk1-Dio3* miRNA cluster. Since miR-494 showed the highest activity in a colony forming unit assay and increased cell cycle progression through MCC targeting, the antitumor potential of miR-494 antagonist molecules was tested in *Myc*-driven HCC bearing mice showing decreased tumor burden associated with increased p27 and MCC protein levels [16]. In line, we demonstrated p27 as a miR-494 direct target gene in HCC cell lines and xenograft mice playing a role in cell cycle fastening and cell proliferation in miR-494 overexpressing Huh-7 cells, suggesting that both animal models might be suitable for identification of miR-494 downstream molecular mechanisms [15]. Similarly, miR-206 was commonly downregulated among *c-Myc* and *AKT*/*Ras*-oncogene induced HCC mouse models, HCC cell lines, and human HCCs contributing to tumor development and progression *through cell cycle regulator cyclin D1* (*CCND1*), *c-MET*, and cyclin-dependent kinase 6 (*CDK6*) direct targeting, as demonstrated by in vitro and in vivo experiments [21]. Notably, oncogene overexpression might be obtained not only through the establishment of TG mice but, more easily, through hydrodynamic tail-vein injection, which is an efficient procedure to deliver nucleic acids, primarily to the liver, taking advantage of its peculiar fenestrated sinusoids. By means of a rapid injection of a large liquid volume containing the specific sequences of interest, mouse models with stable gene expression can be generated. Strikingly, miR-206 hydrodynamic delivery in concomitance with *c-Myc* and *AKT*/*Ras* oncogenes prevented tumor formation and cancer-associated death of all animals, emphasizing the suppressor role for miR-206 in HCC initiation in both oncogene-derived mouse models.

In conclusion, genetically engineered models represent optimal tools for investigating miRNA involvement in hepatocarcinogenesis (Dicer1KO, miR-122KO), for testing anticancer activity of miRNA-based therapeutic option (TG221 + DEN; HBx TG; oncogene-related TG mice) and for recreating the multi-step pathogenic history of HCC development in a cirrhotic (TG221 + CCl4) or metabolic (miR-223KO) background.

## 7. HCC Mouse Models and miRNA-Based Therapeutic Approaches

A growing body of evidence support the involvement of miRNAs in the coordination of complex cellular programs, due to their ability to regulate hundreds or thousands of targets. Since the liver represents an ideal organ for gene therapy, first in vivo studies demonstrated the nontoxicity and efficacy of chemically modified oligonucleotides targeting the liver-specific miR-122 in both rodent and nonhuman primate species [32,62]. Regarding miRNA-based therapeutic strategies, several pros and cons have to be considered when clinical trials are concerned. Indeed, on the one hand, the intrinsic nature of miRNAs as multi-regulators of gene expression represents an advantage for cancer treatment favoring the targeting of multiple steps of a single pathway, as well as contemporaneous inhibition of redundant pathways that could represent tumor escape mechanisms associated with drug-resistant phenotypes. On the other hand, this multitargeting ability might trigger opposite effects, depending on tissue of origin, but also on cellular context within a single tumor, where the same miRNA might behave as a tumor-suppressor gene or as an oncogene, depending on constitutive expression of targets [63]. As an example, we reported the dual role for oncomiR-221 in cell proliferation and drug sensitization based on p53 status, as well as the simultaneous targeting of *PTEN* tumor-suppressor gene and *AKT3* oncogene, both belonging to the AKT/mTOR pathway, by miR-519d, which is strongly upregulated in 50% of HCCs [64,65]. These apparent contradictory functions of miRNAs can be understood when we consider their physiologic activities and their prominent role that is the fine-tuning of cellular processes, balancing complex molecular networks at multiple levels. Of note, this equilibrium might be no more applicable to diseased states such as cancer. Even more importantly, this concept should be considered when we hypothesize to use miRNAs as therapeutic molecules. In this regard, it is mandatory to identify those molecular settings in which miRNA manipulation targets specific driver mechanisms. In the abovementioned case of miR-519d, which targets both *PTEN* and *AKT3*, its inhibition in HCCs with high *PTEN* expression is able to counteract AKT/mTOR pathway. Conversely, in HCC with *PTEN* null expression, it would determine an even stronger AKT/mTOR pathway activation through *AKT3* upregulation, contributing to tumor cell proliferation and invasion.

Due to the frequent downregulation of miR-34a in most of human tumors, including HCC, the first clinical trial testing of a liposomal miR-34a mimics-based formulation is now ongoing in oncologic patients at advanced stages [66]. Notably, Gougelet and colleagues described the overexpression of miR-34a in HBV-related HCCs and APC^−/−^ transgenic mice where its silencing by means of locked nucleic acid (LNA) probes slowed down tumor progression, emphasizing the need for delineation of accurate patients’ subclasses based on their molecular patterns when miRNA-mediate approaches are concerned [22]. In this scenario, miRNAs represent optimal target genes and attractive candidates for single or combined therapeutic strategies. In the abovementioned oncogene-related mouse model injected with *AKT*/*Ras* or *c-Myc* to induce HCC, the therapeutic potential of two different miR-206-based molecules was assayed. The first strategy was based on minicircle episomal DNA vectors, which have a smaller size that enables more efficient delivery and sustained expression in a period of weeks with respect to greater DNA plasmid vectors. The second strategy considered miRNA mimics modified at 2’O-methyl and labeled with four cholesterol groups, to increase the stability and transfection efficiency. Both strategies showed a marked anticancer effect in *c-Myc* mice, decreasing tumor growth and target genes expression, but failed to repress tumor growth of *AKT*/*Ras* mice, perhaps due to their more rapid growth, underlining the importance to test miRNA-based therapeutics in different animal models and suggesting again the need for patient stratification based on genomic and transcriptomic backgrounds [21]. In the following paragraphs, we describe miRNA-based antitumor approaches in xenograft and orthotopic HCC mouse models, focusing on their representativeness of human pathology and translation into the clinics (Figure 1).

## 8. Xenograft Mice

Xenograft models are obtained by subcutaneous injection of tumor cells into immunocompromised mice and have been largely used in preclinical studies due to their easy manipulation. They represent a good tool for a rapid in vivo screening of tumor growth and drug response, even though caution has to be used when translating results into the clinical practice. Indeed, major disadvantages with respect to GEM are (1) the absence of a proper immune response by the host and (2) the lack of an appropriate tumor microenvironment. On the other hand, the advantage of xenograft with respect to GEM models is represented by their easy and rapid establishment, as well as by the employment of human cells that might better reflect human tumor phenotypes and drug response with respect to mouse tumors. The ability of cancer cells to develop a multidrug resistant phenotype represents a major obstacle for both standard chemotherapy and molecular targeted therapy leading to treatment failure and tumor progression and limiting effective chances to cure oncologic patients. The presence of redundant molecular mechanisms favors tumor escape, and targeting single molecules often activates compensatory pathways allowing survival of neoplastic/progenitor cells. Since pharmacologic treatments modify a conspicuous number of signaling cascades, the miRNA-based multiple inhibition of regulatory elements combined with clinically available drugs might be a successful strategy in the field of molecular oncology. Regarding HCC, an interesting study by Tang and coworkers described the antiproliferative effect of an artificial long noncoding RNA (AlncRNA) simultaneously targeting six miRNAs whose expression is upregulated during sorafenib treatment, namely miR-21, miR-153, miR-216a, miR-217, miR-494, and miR-10a-5p. Interestingly, *PTEN* is a common and validated target of all these miRNAs, as well as the principal negative regulator of the pro-survival AKT/mTOR pathway, whose activation represents an unfavorable event often occurring in sorafenib-acquired resistance and promoting cell proliferation and autophagy blockage [67]. Intratumoral injection of adenoviral particles Ad5-AlncRNA in a sorafenib-resistant xenograft mouse model induced only a slight decrease of tumor growth but strongly synergized sorafenib antitumor activity, half-reducing tumor weight, decreasing Ki-67 staining, and increasing terminal deoxynucleotidyl transferase dUTP nick end labeling (TUNEL)-positive cells with respect to control animals [68]. In agreement, in our experience, the only overexpression of miR-494 in HCC xenografts does not impact on tumor growth, whereas in the setting of sorafenib treatment, overexpression of miR-494 determines resistance, suggesting the central role for miR-494/PTEN/AKT/mTOR/p27 axis in sorafenib sensitization in HCC [15]. He and coworkers described similar findings reporting the synergistic effect of anti-miR-21 oligonucleotides coupled with sorafenib in a xenograft model obtained by subcutaneous inoculation of sorafenib resistant Huh-7 cells. In particular, miR-21 silencing in vivo restored sorafenib sensitization through PTEN targeting and autophagy activation as assessed by Ki-67 and TUNEL staining as well as by transmission electron microscopy showing a consistent increase of apoptotic cell death and autophagosome vesicles [23].

In the last years, several efforts were made toward the development of efficient nanotherapeutics for the delivery of miRNA-based oligonucleotides, increasing organ-specific targeting and reducing miRNA degradation by systemic endonucleases. Interestingly, a recent study developed a straightforward strategy to deliver miR-375 in HCC cells in both in vitro and in vivo studies assembling miRNA mimics molecules on the surface of gold nanoparticles (AuNP). Several properties make these oligonucleotide-functionalized nanoparticles ideal for biomedical applications, such as their ability to be captured and internalized by cultured cells or animal tissue without the aid of other carriers, the increased nuclease resistance of coated miRNAs and a lower activation of host immune response. Intratumoral injection of miR-375 nanoparticle formulation markedly reduced tumor growth of HepG2 xenograft masses. These findings were confirmed in a second HCC mouse model obtained by hydrodynamic injection of *AKT*/*Ras* oncogenes and tail-vein administration of AuNP-miR-375 showing a consistent increase of apoptotic markers and a reduction of tumor nodules and hepatocyte proliferation [24].

Several considerations arise from studies on xenograft models: (i) the contemporaneous modulation of different miRNAs targeting common pathways might help control tumor recurrence and drug-resistant phenotypes (lnRNA-mediated approaches); (ii) the combination of miRNAs and chemotherapeutics or targeted molecules might potentiate drug efficacy (miR-494 and miR-21 combined with sorafenib); (iii) the development of delivery systems favors optimal miRNA uptake by cancer cells.

## 9. Patient-Derived Xenograft Mice

The employment of patient-derived xenograft (PDX) mice shows several advantages over cell-line-derived xenografts: (1) maintenance of genetic and epigenetic abnormalities and clonal heterogeneity existing in the primary tumors; (2) stroma cells from the tumor can be implanted, mimicking the tumor microenvironment; and (3) stronger predictive results that might aid in choosing the best therapeutic strategy for the patient.

An interesting study by Hou and collaborators described the results from a quantitative analysis of miRNA abundance in the liver and identified miR-199a/b-3p as the third most highly expressed miRNA resulting in deregulated in almost all HCC cases, independently from etiologic factors. Interestingly, nine miRNAs accounted for 88.2% of the liver miRNAome and specifically miR-122, miR-192, and miR-199a/b-3p represented 52%, 16.9%, and 4.9%, respectively. In addition, epigenetic modifications of histones mediated miR-199a/b-3p downregulation in HCC, which was associated with decreased tumor-free results and overall survival in two patient cohorts, activating the oncogenic Raf/MEK/ERK signaling cascade. Regarding in vivo models, a patient-derived mouse model was established by subcutaneous transplantation of fresh tumor tissue in nude mice and masses propagated by sequential subcutaneous passages. Different miRNA delivery strategies were assayed, consisting of intratumoral injection of cholesterol-conjugated miR-199a/b-3p mimics or AAV vector particles inhibiting tumor growth, decreasing AFP levels, and causing tumor necrosis. In addition, AAV-mediated miR-199a/b-3p delivery by a single tail-vein injection in patient-derived orthotopic mice showed increased miRNA levels in tumor tissue, together with reduced tumor size, demonstrating the efficacy and nontoxicity of AAV8 vector system for potential HCC gene therapy [46]. Findings here obtained with miR-199a-3p-based therapeutics agree with those reported in the TG221 mice, proving the reliability of these animal models and suggesting miR-199a-3p replacement as a promising therapeutic option for human HCC [19].

In this scenario, the targeting of multiple oncogenic miRNAs by lncRNA-based vector engineering is emerging as a winning and innovative therapeutic strategy avoiding cancer cells to bypass the inhibition of a single miRNA and regain their proliferation capabilities. Li and coworkers generated an artificially designed interfering long noncoding RNA (lncRNAi), which contains the complementary sequences to multiple oncomiRs, competing with their target genes to bind to and consume oncomiRs inside HCC cells. Among oncomiRs overexpressed in HCC, this lncRNAi oncolytic vector was designed to selectively target miR-21, miR-221/222, miR-224, miR17-5p/20a, miR-10b, miR-106b, miR-151-5p, miR-155, miR-181a/181b, miR-184, miR-1, and miR-501-5p and showed multiple effects on transduction pathways, playing a key role in proliferation, invasion, and apoptosis of HCC cells. Two animal models were employed in this study (xenograft and PDX mice), one receiving intratumoral injection of lncRNAi oncolytic vectors and the other control vectors. A significant decrease of tumor growth was observed in both models, showing a decrease of target miRNAs and a correspondent increase of the most representative target genes, *p27* and *PTEN*, highlighting this multi-miRNAs-based approach as a promising strategy with a greater anticancer efficacy in HCC [69].

## 10. Orthotopic Mice

Considering the importance of tumor microenvironment or tumor ‘niche’ for malignant transformation of normal hepatocytes and for development and progression of tumor nodules, the employment of orthotopic xenografts should be favored when testing drugs acting on tumor microenvironment. An innovative in vivo bioengineering approach has been established for the development of miRNA-based cancer therapies, producing a panel of biologic miRNA/siRNA molecules containing a common tRNA/pre-miR-34a molecular scaffold expressed at high levels by *Escherichia coli* fermentation. The production of these oligonucleotides in living cells guarantees minimal post-transcriptional modifications with respect to heavy chemical modifications applied to oligonucleotides produced by chemical synthesis; moreover, their correct processing and specific activity have been reported in human cells [70]. Interestingly, bioengineered let-7c displayed the highest antiproliferative activity against Huh-7 cells and reduced protein levels of validated targets, such as *LIN28B*, *c-Myc*, AT-rich interactive domain-containing protein 3B (*ARID3B*), and *Bcl-xl*, in both Hep3B and Huh-7 cells, as well as stem-cell properties of Huh-7 cells. An orthotopic HCC xenograft model was employed to test in vivo the efficacy of a lypopoliplex (LPP)-let-7c formulation, displaying a significant reduction of tumor burden associated with decreased AFP serum levels and increased apoptotic markers. LPP-formulated let-7c nanotherapeutics were well tolerated in vivo, showing minimal immunogenicity in human peripheral blood mononuclear cells and immunocompetent mice and holding great promise for the development of miRNA-based therapeutic options entering clinical experimentation [25].

A crucial aspect regarding gene replacement therapy is represented by the abundance of gene expression in normal tissue, reducing the risk of side effects, favoring treatment tolerance, and restricting antiproliferative and proapoptotic effects to cancer cells. miR-26 is expressed at high levels in several tissues but is downregulated in liver tumors, altering cell proliferation of HCC cells through direct targeting of cyclins D2 (*CCND2*) and E2 (*CCNE2*). The antitumorigenic effect of AAV-mediated miR-26 replacement strategy was previously reported in a *c-MYC* transgenic model of liver cancer, resulting in inhibition of cancer cell proliferation and tumor progression [26]. In addition, Jin et al. reported that miR-26 restoration is able to promote doxorubicin sensitization through autophagy inhibition by direct targeting of Unc-51 like autophagy activating kinase 1 (*ULK1*) gene that is responsible for autophagosome enucleation regulating initial steps of autophagic program. In particular, an orthotopic mouse model was established by implanting HepG2 cells in livers of nude mice, showing that miR-26 was able to increase drug sensitivity by blocking autophagy and therefore reducing tumor cell proliferation and triggering apoptotic cell death [27]. Notably, miR-26 family members were also downregulated in our DEN-HCC rat model (miR-26a-5p and miR-26b-5p; raw data are available in ArrayExpress repository, accession number E-MTAB-7624), again highlighting the similarities between rodent and human HCC biology and emphasizing the importance of a thoughtful animal model choice when translational studies are concerned, privileging studies with at least two different preclinical models.

## 11. Rat Models

HCC rat models do not allow genetic manipulation as mouse models do. However, execution of specific treatments such as transcatheter arterial chemoembolization (TACE), bile duct ligation, and surgical procedures are easier in rats than in mice [71]. Same applies to noninvasive monitoring of HCC development by ultrasound (US). HCC rat models display a high analogy to histopathological progression of human HCC and are of great value to identify effective anticancer treatments. Here, we describe the most characterized rat models in terms of molecular match with human HCC and miRNA deregulation: the DEN-HCC and the resistant-hepatocyte (R-H) rat models. We also report data about cirrhotic rats developing HCC, which were investigated to test miRNA-based therapeutic approaches.

## 12. Chemically Induced Rat Model

Due to the essential role of the liver in metabolism and detoxification, several chemical agents inducing liver injury are used as HCC promoters. DEN is a chemical carcinogen with the potential to cause tumors in various organs, including the liver, gastrointestinal tract, and respiratory tract [72]. DEN causes genomic mutations by inducing ethyl adducts and alkylation at many DNA positions [73]. Initiation of carcinogenesis, promotion, and progression can be produced by modulating DEN dosage [72]; in fact, it can induce benign and malignant liver lesions in rats with a high success and a low mortality rate, providing a model that simulates the lesions occurring in human liver. Induction of HCCs is usually obtained by administering DEN in the drinking water (100 mg/L) for eight weeks and one month after the last day of DEN administration multiple nodules can be observed with a size ranging from 2 to 10 mm in diameter [74,75]. By using this approach, we obtained rat HCCs sharing histopathological features of human HCC (Figure 2). Other protocols of DEN administration, such as injection (30 mg/kg body weight twice a week for 11 weeks), followed by a further nine weeks of observation, were also used, and they were reported to reproduce the inflammation-fibrosis-cancer stepwise histopathological progression similar to human HCC [76].

DEN models have been used in many studies aiming at HCC characterization. Next-generation sequencing (NGS) technologies provided new opportunities to identify HCC molecular signatures contributing to treatment optimization. In this context, we compared transcriptomic data from human and DEN-induced rat HCCs. Among common deregulated genes, the activation of Notch pathway and the aberrant expression of a panel of HCC-associated miRNAs (e.g., miR-122, miR-221, and let-7a) suggested this model to be suitable to explore miRNA contribution to disease progression and treatment response [77,78,79]. Remarkably, most of miRNAs deregulated in the DEN-induced rat model are aberrantly expressed in human HCC cohorts from several studies, as summarized in Table 1 and Table 2. In particular, members of both paralogous clusters, miR-17-92 and miR-25-106b, showing oncogenic properties [54], were upregulated in our DEN-HCC rat model (miR-18a-5p and miR-106b-5p; raw data are available in ArrayExpress repository, accession number E-MTAB-7624), as well as in our HCC patient datasets (miR-18a and miR-25) [3]. Similarly, miR-122 was downregulated in 70% of rat DEN-HCCs (unpublished data from our group) and in the same percentage of human HCCs [3]. Let-7 miRNA family, miR-26a, and miR-145 were also downregulated in the rat model mirroring human HCC (Table 1 and Table 2).

The DEN-induced HCC rat model was also used to examine pathways involved in sorafenib response, showing the modulation of apoptotic and proliferative pathways, and to test the therapeutic role of siRNAs/miRNAs. In particular, miRNAs are more attractive than siRNAs due to their simultaneous effects on different pathways. Interestingly, compound Astragalus and Salvia miltiorrhiza extract (CASE) containing active principles extracted from *Astragalus membranaceus* and *Salvia miltiorrhiza* provided anticancer therapy more likely through miR-145 and miR-21 upregulation and downregulation, respectively [166].

miR-494 is overexpressed in the great majority of rat DEN-induced HCCs, and its expression correlates with stem-cell features, representing an appropriate model for studying molecular alterations occurring in the stem-cell-like HCC subgroup with miR-494 aberrant expression and poor survival (25% of human HCCs) [15]. Pollutri et al. also demonstrated an antitumor effect of a sorafenib-combined anti-miR-494 strategy in DEN HCC rats, showing decreased tumor progression with respect to sorafenib monotherapy. miR-494-mediated mTOR pathway activation was responsible for decreased sorafenib sensitization in HCC preclinical models; indeed, rapamycin co-administration triggered apoptotic cell death in miR-494 overexpressing HCC cells, as evidenced by Poly adenosine diphosphate [ADP-ribose] polymerase 1 (PARP-1) and caspase-3 specific cleavage. Remarkably, all these findings were replicated in xenograft mice, emphasizing the usefulness of both models for HCC preclinical studies. In the same model, miR-221 overexpression associated with sorafenib resistance and caspase-3 was recognized as its target gene, contributing to miR-221 anti-apoptotic activity and drug-resistant phenotype. Notably, an association between high miR-221 or low caspase-3 levels and tumor multifocality was assessed in human HCC specimens, suggesting the robustness and specificity of preclinical data and their reliability with respect to the human pathology [167].

Chemically induced models are advantageous to study HCCs developed in a natural background of liver damage. However, age of the animals and hepatotoxins concentration might lead to different tumor phenotypes, influencing experimental outcomes. Remarkably, DEN-induced liver tumors were responsive to sorafenib treatment, whereas orthotopically implanted rat HCCs were not [168]. These findings point out differences in treatment responses between models that might be related to tumor microenvironment and might have a translational impact. As observed for mouse models, the relevance of cirrhosis in the natural history of human HCC development has prompted researchers to recreate this background in laboratory models, too. Experimental approaches used to induce liver damage progressing to cirrhosis in rats include heterogeneous protocols of exposure to CCl4, DEN, bile duct ligation, or a combination of these [169]. Some studies performed in HCC rat models arising on cirrhosis have specifically addressed the role of miRNAs as possible treatments for helping liver regeneration of non-neoplastic hepatocytes after partial hepatectomy, without triggering proliferation of residual HCC cells. Indeed, the impaired regenerative ability of cirrhotic liver following liver resection still represents an important risk factor for post-hepatectomy liver failure. In the study by Chen and coworkers, liver cirrhosis was induced by intraperitoneal injection of 50% CCl4-olive-oil solution in adult male Wistar rats [170]. In a first study conducted in cirrhotic rats undergoing partial hepatectomy, miR-203 mimics enhanced proliferative signals promoting liver regeneration such as IL6/STAT3, favoring the proliferation of normal hepatocytes. Even more interestingly, the same research group further demonstrated a dual role of miR-203 by addressing the problem of HCC control after partial hepatectomy, which may trigger growth and metastasis of residual HCC cells. In particular, Zheng et al. induced cirrhosis and HCC through a sequential exposure of CCl4 and DEN, reproducing progression of HCC in human cirrhotic livers [171]. These Authors confirmed miR-203 as an enhancer of liver regeneration after 70% hepatectomy through targeting of IL-6/SOCS3/STAT3 signaling. At the same time, they demonstrated a tumor suppressive effect of miR-203 on HCC cells. Indeed, miR-203 mimics inhibited proliferation, invasion, and lung metastasis of residual HCC cells by suppressing epithelial-to-mesenchymal transition (EMT) through IL-1b, Snail1, and Twist1 targeting. The dual role of miR-203 is not so unexpected; indeed, the same miRNA may act in opposite ways not only in different tissues, but also in the same tissue, depending on the constitutive expression of targets.

## 13. R-H Rat Model

Hepatocarcinogenesis is a multistep process characterized by the progressive accumulation of genetic mutations, epigenetic changes, and chromosomal rearrangements that, together with chromatin instability, give rise to hepatocyte transformation and the onset of heterogeneous tumors. Because of limitations associated with tissue availability at early stages, animal models that fully recapitulate HCC history represent valuable tools for comprehensive studies on molecular alterations involved in its early development. The resistant-hepatocyte (R-H) rat model consists of a single carcinogenic dose of DEN followed by a short-term dietary exposure to 2-acetylaminofluorene (2-AAF) that suppresses growth of virtually all normal hepatocytes, combined to partial hepatectomy (2/3 of rat liver) to sharpen the rapid proliferation of DEN-altered hepatocytes. After one week from partial hepatectomy, hepatic foci will continue to proliferate, becoming the precursor lesions that, only in part, will evolve into HCC nodules [172]. This rat model represents an ideal preclinical tool for the study of molecular events occurring at very early stages of hepatocarcinogenesis, allowing the identification of phenotypically distinct lesions that can be easily isolated by means of laser capture microdissection (LCM). The evolution of preneoplastic lesions has been characterized as showing a positivity for the hepatocyte progenitor cell marker, cytokeratin-19 (CK19), in 25% of preneoplastic nodules, in 50% of persistent lesions, and in all HCCs, demonstrating that CK19-positive preneoplastic lesions are more likely the progenitor of HCC in this model. Comparative functional genomics showed that CK19-associated gene signature could predict clinical outcomes of HCC patients, representing an ideal model for the study of the HPC-derived HCC subclass [173]. miRNA and gene-expression profiling identified the activation of nuclear factor erythroid related factor 2 (NRF2) pathway and upregulation of miR-200a as the prevalent molecular alterations occurring at very early stages of tumorigenesis and maintained throughout the carcinogenic process. NRF2 is an integrated redox signaling system controlled by Kelch-like ECH-associated protein 1 (*KEAP1*), which promotes its proteasome-mediated degradation and whose regulation is modulated by miR-200a itself. The association between miR-200a/KEAP1/NRF2 axis and tumorigenic potential was proven by in vivo experiments, showing that NFR2 pathway is turned off during tumor regression of HCC nodules. Because of the high concordance between altered genes in rat and human tumors, the R-H model also represents an ideal tool for the detection of miRNA alterations occurring at very early stages of cancer development and for the identification of target genes and innovative therapeutic strategies [174]. In addition, the activation of NRF2 pathway is also responsible for the upregulation of glucose-6-phosphate dehydrogenase (*G6PD*), a key enzyme implicated in metabolic shift or ‘Warburg effect’, through miR-1 inhibition [175]. High *G6PD* expression levels were detected in CK19-positive lesions and early preneoplastic foci, but not in CK-19-negative lesions, and, in line, increased *G6PD* mRNA levels were observed in two HCC patient cohorts where it positively correlated with CK19 expression, higher tumor grade, and increased metastasis formation. In agreement, a negative correlation between miR-1 and *G6PD* was identified in CK19-positive lesions, and lower miR-1 levels were quantified in human HCCs with respect to surrounding livers, confirming the high concordance in terms of dysregulated pathways between aggressive preneoplastic rat lesions and human HCCs [176]. Interestingly, a global proteomic analysis from glutathione S-transferase-P (GST-P) positive laser micro-dissected nodules identified *G6PD* among protein markers discriminating R-H model-derived focal lesion from normal liver with or without progenitor cell activation, demonstrating LCM coupled with mass-spectrometry-based proteomics as an effective approach to characterize preneoplastic lesions and to identify early diagnostic markers for effective clinical intervention. In addition, mTOR pathway was found activated at early stages of carcinogenesis and, consistently, its transient inhibition by rapamycin treatment impaired the growth of focal lesions and induced a less aggressive phenotype attenuating the loss of differentiating functions, as detected by both transcriptomic and proteomic genome-wide analyses [177,178]. The AKT/mTOR pathway is frequently activated in HCC, characterizing a subgroup of patients with a high proliferation signature and sorafenib resistance [179,180,181]. Several miRNAs contribute to its post-transcriptional regulation mediating sorafenib resistance in preclinical models, highlighting its central role in HCC progression and treatment escape [11,15,68,114,167]. Interestingly, the R-H model is also suitable for the investigation of oval cell population, due to its rapid expansion following partial hepatectomy. The Hippo pathway is required for the repression of oval cells that represent the adult liver’s stem-cell compartment. An accumulation of *YAP1*, a transcriptional co-activator of the Hippo pathway, was detected in preneoplastic foci, representing a very early event in hepatocarcinogenesis. On the contrary, a decrease of miR-375 was found in rat early lesions, suggesting its contribution to *YAP1* accumulation and HCC development and confirming its role as a TS miRNA in HCC [182]. Remarkably, the downregulation of thyroid hormone receptor beta (*THRB*) was identified as an early event in the R-H model, but it was not associated with genetic mutations or DNA methylation of its promoter region. On the other end, among miRNAs directly targeting *THRB* mRNA, miR-27a, miR-181a, miR-146a, and miR-204 were upregulated in rat HCCs, and, in particular, miR-27a showed an inverse correlation with *THRB* levels and inhibited its expression in HCC cell lines. Regarding human HCC, reduced expression of *THRB* target genes, iodothyronine deiodinase 1 (*DIO1*) and *G6PC*, was confirmed in cirrhotic samples with respect to normal livers, suggesting the alteration of this pathway as an early event in human HCCs, as well. In agreement, an upregulation of miR-181a, miR-27, and miR-204 was detected in surrounding cirrhotic livers, more likely contributing to *THRB* modulation in liver cirrhosis and increasing HCC risk development due to the so-called ‘field effect’, further strengthening the similarities between the RH-model and the human counterpart [183]. Similarly, Brockhausen and coworker reported the increased expression of miR-181a in human and rodent cirrhotic livers when compared to normal livers and demonstrated its involvement in TGF-β-mediated EMT in immortalized non-neoplastic human hepatocytes [184]. TGF-β is a signaling pathway strongly activated in chronic liver disease, further supporting the possible role for miR-181a in neoplastic transformation in a milieu of chronic inflammation [184,185]. In addition, lncRNA CCAT1 was demonstrated among molecular mechanisms contributing to miR-181a deregulation through a sponging activity leading to increased cell proliferation and ATG7-mediated autophagy of HCC cells [186]. Concerning autophagy, the R-H rat model was employed to investigate whether its dysregulation might occur at early stages of liver carcinogenesis. Controversial data report the dual role of autophagy in cancer, acting both as a cell survival process and as a tumor suppressor pathway, likely suppressing tumor growth at early stages and favoring cell proliferation and energy supply at later stages [187]. The study by Kowalik and coauthors reported an impairment of autophagy in preneoplastic lesions characterized by accumulation of abnormal mitochondria and autophagic vacuoles with partially degraded material, displaying an early deregulation of autophagy during hepatocarcinogenesis. An increase of autophagic markers, Ambra1, Beclin1, and p62 was detected in CK19-positive nodules only. Strikingly, a 150-fold upregulation of the oncogenic autophagy-related miRNA, miR-224, was detected in these lesions, and its accumulation is more likely associated with the deregulation of autophagic machinery. In vivo experiments with amiodarone, an autophagic inducer, showed augmented tumor progression with increased size of CK19-positive lesions, and, vice versa, treatment with the autophagy inhibitor chloroquine reduced size and percentage of CK19-positive nodules [188]. Due to the contradictory role of autophagy in tumorigenesis, particular attention should be given when a pharmacological strategy is concerned, considering the employment of different animal models to better guarantee the translation of preclinical findings into the clinics. Others and our group compared this model with the rat DEN HCCs and described similar alterations, which were confirmed in patient cohorts, as well, emphasizing the robustness of these preclinical tools. Specifically, a prominent downregulation of the 2-hydroxy acid oxygenase (*HAO2*) was observed in the R-H rat model at both early and advanced stages of carcinogenesis. *HAO2* was significantly downregulated in 100% of animals from two other HCC rat models, as well as in two mouse models of HCC (with and without DEN administration), and in human HCCs, demonstrating *HAO2* repression as a general mechanism contributing to liver cancer development more likely due to reduced reactive oxygen species levels in transformed cells [189]. A time-course gene-expression profile for Aldo-keto reductase (*Akr*) family members in preneoplastic and tumor lesions identified the upregulation of six of its members in the R-H rat model, representing possible markers of early diagnosis. This enzyme superfamily contributes to the metabolism of steroids, carbohydrates, prostaglandins, aldehydes, and ketones, and their high representation in HCC has been previously reported by proteomic investigations [190]. The overexpression of two out of six members was confirmed in DEN-HCC rats and in human liver biopsies at both mRNA and protein levels, and, as mentioned above, *NRF2* nuclear translocation was detected in tumor samples, confirming the central role of this transcription factor in promoting liver cancer [191]. In conclusion, the R-H rat model represents a helpful tool for the discovery of dysregulated events occurring at early stages of tumorigenesis, showing a high translational potential because of great representativeness of molecular pathways altered in human HCC.

## 14. Zebrafish

Although zebrafish are less complex than humans, their liver is similar in structure to a mammalian liver, with biliary epithelial cells, sinusoidal endothelial cells, and hepatic stellate cells that support hepatocytes [192]. A zebrafish liver develops HCC similar to human liver, and it can be used to study putative driver mutations found in HCC. Zebrafish and human HCCs share overlapping expression profiles of genes involved in apoptosis, DNA replication, cell adhesion, metastasis, cytoskeletal organization, cell motility, cell cycle/proliferation, RNA processing, and protein synthesis. Thus, zebrafish liver tumors are highly analogous to human HCC in terms of comparative analyses of microarray data [193]. Upregulated and downregulated genes in zebrafish liver tumors are also constantly upregulated and downregulated in human HCCs [194]. MicroRNAs are known to be evolutionary conserved across species. Among them, miR-122 was found to be conserved in 12 different species, including humans, frogs, and zebrafish, and it is involved both in human and zebrafish hepatocytes differentiation [195,196]. Moreover, several putative target genes of the miR-1, miR-146, and miR-221/miR-222 families are conserved between human and zebrafish. Hepatocyte differentiation is directed by a positive-feedback loop that includes the transcription factor (HNF6) and microRNA (miR-122), both in human and in zebrafish liver [197]. According to its tumor-suppressor function in human HCC, miR-145 expression regulates zebrafish embryonic liver size by controlling hepatocyte proliferation [198]. Functional analysis in zebrafish shows that miR-30a is required for normal hepatobiliary development, suggesting a similar critical role in mammalian hepatogenesis [199]. The conserved expression signatures of these miRNAs and the involvement of their predicted target genes in cancer support zebrafish as a model of liver cancer representative of the human counterpart.

Following injection of the human HCC cell line JHH6 into the yolk sack of Zebrafish larvae, Tonon et al. confirmed this model as a useful setting to test therapeutic molecules for HCC [200]. Moreover, injecting human cancer cells into zebrafish embryos when the immune response is not yet established, prevents their rejection. More recently, small molecules with strong anticancer therapeutic potential and better therapeutic index than sorafenib were identified by using a zebrafish drug-screening platform [201]. As reported in others HCC animal models, the role of metformin on HCC tumor surveillance was also demonstrated [202]. Interestingly, as observed in humans, male fish developed HCC faster than females due to sex hormone imbalance via noncanonical estrogen signaling, including G protein–coupled estrogen receptor 1 (GPER) [203]. In line, fish treated with 17 β-estradiol developed more tumors than controls, and this effect was evident in particular in male cohorts [204]. Together with transgenic technology, several inducible expression systems are available for zebrafish. Among them, a transgenic model expressing kRAS develops liver tumors and appears to be representative of those human HCCs activating Ras signaling, which account for up to 50% of cases [205,206]. Transgenic expression of HBx in a TP53 mutant background induces HCC in 44% of fish, whereas zebrafish expressing HCV core protein swiftly develop HCC when exposed to carcinogens [207,208]. Up to 40% of human HCCs result from activating mutations in the gene encoding for β-catenin (CTNNB1). The zebrafish model was used to prove that activated β-catenin expression in a small subset of hepatocytes is sufficient to drive HCC initiation [209]. Zebrafish are effective as disease models because of their high fecundity, low housing cost, and transparency, which allows live imaging and is useful to explore mechanisms by which HBV and HCV induce HCC. Moreover, large experimental sample sizes are allowed, allowing informative studies. Finally, small molecules can be added directly to the zebrafish’s water, thus avoiding the gavage often required in rodents. The disadvantages are mainly related to the low amounts of liver tissue available for different assays and the lack of antibodies able to recognize zebrafish antigens.

## 15. Conclusions

Advances in tumor biology have led to the identification of new targets that hold promise for ameliorating HCC patients’ outcome. Translation of novel treatments requires preclinical testing in animal models mirroring human HCC. New approaches should be tested in specific models, identified on the basis of molecular matching. A wide range of rodent models of HCC is now available. Most of them were characterized from a molecular point of view, including miRNA aberrant expression. Remarkably, the multitargeting function of miRNAs make it mandatory to define the molecular context for each experimental purpose and to fully characterize each model not only for miRNA but also for targets expression, at least as far as the oncogenic drivers are concerned. Indeed, targeted treatments should be tested in models that are representative of specific HCC subgroups, in order to assess those settings in which they are awaited to work at their best.

The restoration of miR-199a-3p and miR-26a levels as promising miRNA-based anticancer strategies have been assayed in several HCC animal models by using different therapeutic approaches, such as miRNA mimics or AAV particles. On the other side, the silencing of onco-miR-221 holds great promise as a therapeutic option in the oncologic field. Indeed, miR-221 plays oncologic functions in almost all solid malignancies, and its inhibition represents a possible common strategy for different cancer types. Interestingly, the development of innovative strategies for the production of miRNA-modified molecules with a nonimmunogenic backbone to increase the stability in body fluids, and the engineering of nanoparticles for enhanced miRNA delivery, are further steps toward the translation of preclinical findings into the clinics.

Besides rodents, other models appear very interesting, such as, for example, zebrafish, which are easier to house, but they are still rarely adopted for preclinical studies on HCC.

In conclusion, HCC models focused to miRNA studies should be chosen based on a profound knowledge of the molecular context, with specific reference to expression assessment of oncogenic drivers targeted by the miRNA of interest. In addition, in HCC several deregulated miRNAs share common targets. Several oncogenic drivers are targeted by multiple deregulated miRNAs. This opens the door to combined approaches aimed at modulation of panels of critical miRNAs, instead of single miRNA. These combined approaches might help to control the well-known feedback loops elicited by miRNAs’ enforced expression or inhibition, which may impair the success of miRNAs manipulation. Indeed, the complex action miRNAs, the modulation of many targets, sometimes with opposite functions, should always be kept in mind, in order to avoid unexpected results in the translational steps.

## Figures and Tables

**Figure 1 cancers-11-01906-f001:**
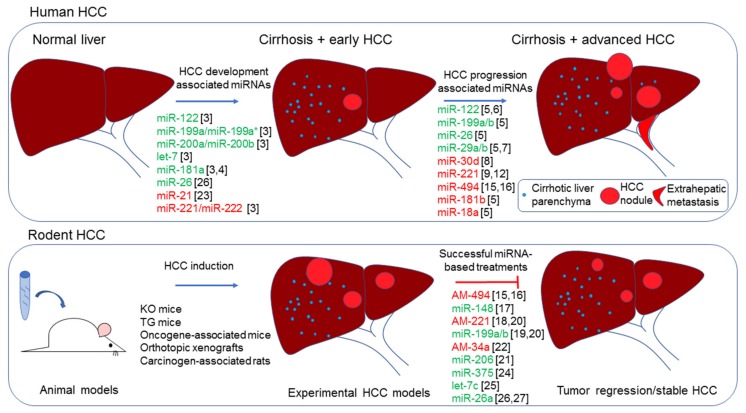
miRNA deregulated in human and rodent human hepatocellular carcinomas (HCCs) and miRNA-based approaches in preclinical models. Top panel depicts miRNA involved in human hepatocarcinogenesis from early development to advanced stages. Downregulated miRNAs are in green; upregulated miRNAs are in red. Bottom panel depicts experimental animal models of HCC and successful miRNA treatments reducing tumor burden. miRNA mimics treatments are in green; anti-miRNA (AM) treatments are in red [3,4,5,6,7,8,9,12,15,16,17,18,19,20,21,22,23,24,25,26,27].

**Figure 2 cancers-11-01906-f002:**
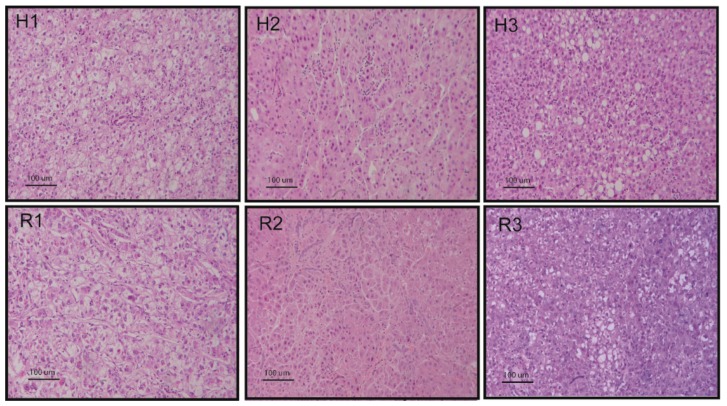
Features of human and rat hepatocellular carcinoma. Hematoxylin-eosin staining of human (**H1–H3**) and rat (**R1–R3**) HCC showing common histopathological features. (H1–R1) Clear cell variant caused by glycogen or lipid accumulation. (H2–R2) Trabecular growth pattern. Neoplastic hepatocytes, with nuclear irregularity, are arranged in trabeculae larger than three cell plates. (H3–R3) Progression toward less-differentiated histopathological grades is marked by the appearance of irregular nuclei, nuclear, and cellular pleomorphism, frequent mitosis, and loss of trabecular or pseudoglandular structures. Original magnification 40×. Scale bars: 100 µm.

**Table 1 cancers-11-01906-t001:** miRNA downregulated in the Diethylnitrosamine-Hepatocellular carcinoma (DEN-HCC) rat model.

Rat Deregulated miRNA	Fold Change (HCC/Surrounding Liver)	*p*-Value	Deregulation in Human HCC	Target Genes in Human HCC	References
rno-miR-378a-5p	18.3	<0.0001	NA	NA	NA
rno-miR-22-5p	11.9	0.012	NA	NA	NA
rno-miR-338-3p	11.5	0.0017	DOWN; DOWN; NA	NA; *HIF1A*; *PKLR*	[80,81,82]
rno-miR-455-3p	10.1	0.0011	DOWN	*STK17B*	[83]
rno-miR-144-3p	8.7	0.0011	DOWN; DOWN	*SMAD4; CCNB1*	[84,85]
rno-miR-203a-3p	7.7	0.014	DOWN	*GPC3*	[86]
rno-miR-352	6.3	0.0098	Rodent-specific miRNA		
rno-miR-30b-3p	6.2	0.0040	NA	NA	NA
rno-miR-10b-5p	5.6	0.012	UP; UP	*HOXD10*; *CSMD1*	[87,88]
rno-miR-99a-3p	5.6	0.0016	NA	NA	NA
rno-miR-210-3p	5.5	0.0083	UP; UP	*VMP1*; *HIF3A*	[89,90]
rno-miR-362-3p	5.1	0.0083	UP; DOWN	*TOB2*; *RAB23*	[91,92]
rno-miR-30e-3p	5.0	0.0061	NA	NA	NA
rno-miR-511-3p	4.7	0.0083	DOWN; DOWN	NA; *AKT1*	[93,94]
rno-miR-193-5p	4.3	<0.0001	DOWN; DOWN	NA; *SPOCK1*	[95,96]
rno-miR-33-5p	4.1	0.0021	NA	NA	NA
rno-miR-192-3p	4.1	0.0056	NA	NA	NA
rno-miR-3547	3.7	0.0059	Not annotated human miRNA		
rno-miR-145-5p	3.7	0.011	DOWN; DOWN; DOWN; DOWN	NA; *IRS1*; *IRS2*; NA	[3,58,97,98]
rno-miR-378b	3.5	0.0039	NA	NA	NA
rno-miR-122-3p	3.1	0.0068	DOWN	*MDM2*	[99]
rno-miR-339-3p	3.0	0.016	NA	NA	NA
rno-miR-30a-3p	3.0	0.012	DOWN	NA	[100]
rno-miR-101a-3p	2.6	0.0010	DOWN; DOWN; DOWN	*MCl-1*; *SOX9*; *EZH2*	[101,102,103]
rno-miR-193-3p	2.4	0.0011	DOWN; DOWN; NA	*uPA*; NA; *CCND1*	[95,104,105]
rno-miR-365-3p	2.2	0.013	DOWN; DOWN; DOWN	NA; *ADAM10*; *RAC1*	[106,107,108]
rno-miR-347	2.2	0.0047	Not annotated human miRNA		
rno-miR-7a-1-3p	2.1	0.017	Not annotated human miRNA		
rno-miR-125b-5p	2.1	0.0044	DOWN; DOWN; DOWN; DOWN; DOWN	NA; *LIN28B*; *PlGF*; *Bcl-2*; NA	[109,110,111,112,113]
rno-miR-122-5p	2.1	<0.0001	DOWN; DOWN	*CGN1*; *ADAM10*, *SRF*, *Igf1R*	[3,114]
rno-miR-140-3p	2.0	0.016	NA	*GRN*	[115]
rno-miR-101b-3p	1.9	0.0054	DOWN; DOWN; DOWN	*MCl-1*; *SOX9*; *EZH2*	[101,102,103]
rno-miR-10a-5p	1.8	0.015	UP; NA	*EphA4*; *NCOR2*	[116,117]
rno-miR-22-3p	1.8	0.0041	DOWN; DOWN; DOWN	*HDAC4*; *YWHAZ*; *CD147*	[118,119,120]
rno-miR-26b-5p	1.8	<0.0001	DOWN; DOWN; DOWN; DOWN	*CDK6*, *CCNE1*; *USP9X*; *SMAD1*; *ULK1*	[27,121,122,123]
rno-miR-26a-5p	1.7	0.013	DOWN; DOWN; DOWN; DOWN; DOWN; DOWN	*CCND2*, *CCNE2*; *CDK6*, *CCNE1*; *IL6*; *VEGFA*; *ITGA5*; *ULK1*	[26,27,121,124,125,126]
rno-miR-99a-5p	1.7	0.0013	DOWN; DOWN; DOWN	*MTOR*, *IGF-1R*; *PLK1*; *AGO2*	[127,128,129]
rno-miR-451-5p	1.6	0.011	DOWN; DOWN; DOWN; DOWN	*IKBKB*; *ATF2*; *c-Myc*; *YWHAZ*	[130,131,132,133]
rno-miR-30e-5p	1.6	0.003	DOWN	*MTA1*	[134]
rno-miR-192-5p	1.6	<0.0001	DOWN; DOWN; DOWN	*ZEB1*; *SLC39A6*; *HOTTIP*	[135,136,137]
rno-miR-140-5p	1.6	0.0028	DOWN; DOWN; DOWN	*TGFBR1*, *FGF9*; *Pin1*; *FEN1*	[138,139,140]
rno-miR-345-5p	1.5	0.0046	DOWN; DOWN	*IRF1*; *YAP1*	[141,142]
rno-miR-30d-5p	1.5	0.0097	UP; UP	*GNAI2*; *GLDC*	[8,143]

NA: data not available; UP: upregulated miRNAs in human cohorts; DOWN: downregulated miRNAs in human cohorts.

**Table 2 cancers-11-01906-t002:** miRNA upregulated in the DEN-HCC rat model.

Rat Deregulated miRNA	Fold Change (HCC/Surrounding Liver)	*p*-Value	Deregulation in Human HCC	Target Genes in Human HCC	References
rno-miR-146b-5p	55.6	<0.0001	DOWN; DOWN	*TRAF6*; NA	[3,144]
rno-miR-183-5p	45.4	<0.0001	UP; UP	*PDCD4*; *FOXO1*	[145,146]
rno-miR-18a-5p	14.2	<0.0001	UP; UP; UP	*ESR1*; *KLF4*; *SOCS5*	[147,148,149]
rno-miR-211-3p	12.9	0.0073	NA	NA	NA
rno-miR-125a-3p	10.8	0.0059	NA	NA	NA
rno-miR-182	8.3	<0.0001	UP; UP	*MTSS1*; *FOXO1*	[146,150]
rno-miR-1249	5.3	0.018	NA	*HNRNPK*	[151]
rno-miR-3473	4.1	0.0050	Rodent-specific family		
rno-miR-466b-5p	3.2	0.016	Rodent-specific family		
rno-miR-466c-3p	3.2	0.043	Rodent-specific family		
rno-miR-1949	3.0	<0.0001	Rodent-specific family		
rno-miR-96-5p	2.7	<0.0001	UP; UP	NA; *FOXO1*	[12,146]
rno-miR-15b-5p	2.0	<0.0001	UP; UP; UP	NA; *Rab1A*; NA	[152,153,154]
rno-miR-93-5p	1.9	<0.0001	UP; UP; UP	NA; *PDCD4*; NA	[12,155,156]
rno-miR-130a-3p	1.8	<0.0001	DOWN; DOWN	NA; NA	[3,157]
rno-miR-106b-5p	1.8	<0.0001	UP; UP; UP	NA; *DR4*; *PTEN*	[158,159,160]
rno-miR-25-3p	1.7	<0.0001	UP; UP	*RhoGDI-1*; *SOCS5*	[149,161]
rno-miR-31a-5p	1.7	0.0076	UP; DOWN	NA; *HDAC2*, *CDK2*	[162,163]
rno-miR-29b-3p	1.5	0.023	DOWN; DOWN	*Bcl-2*, *Mcl-1*; *LOXL2*	[164,165]

NA: data not available; UP: upregulated miRNAs in human cohorts; DOWN: downregulated miRNAs in human cohorts.

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
