# Peer review of "MicroRNAs in Animal Models of HCC"

_cancers, 2019, doi:10.3390/cancers11121906_

Round 1

Reviewer 1 Report

In their review “MicroRNAs in animal models of HCC“, Fornari and colleagues give an extensive overview of the various roles microRNAs (miRNAs) play in animal models of hepatocellular carcinoma (HCC). The authors discuss miRNA knockout and transgenic mouse models, oncogene-related transgenic mice, HCC mouse models suitable for testing miRNA-based therapeutic approaches, xenograft- and patient-derived xenograft mice as well as orthotopic mice. As non-murine models for HCC, several rat models as well as zebrafish models are described. The review is well written and I enjoyed reading this overview. Some comments as mentioned below.

Comments:

+ I would suggest to increase the font size of the mentioned miRNAs in Fig.1, as it was (for me) quite difficult to read, even more so on the print out.

+ I am well aware that a lot of research on HCC and the influence of miRNAs on disease etiology was performed by researchers around Prof. Negrini’s group. However, I would kindly ask to consider a balanced choice of referenced publications. These could include, amongst others, https://doi.org/10.1186/s12885-017-3860-x, https://doi.org/10.1038/s41598-018-30682-5, https://doi.org/10.18632/oncotarget.4202,...

+ p3., line 101: I would use “germ line” instead of “germinal” (if I understood the sentence correctly).

+ Although work on HCC in Zebrafish is probably not so extensive, I would try to slightly extend this chapter. As is, this chapter reads a bit as if it was artificially put together from several unrelated paragraphs, and some of them contain errors (e.g. the reference on p. 19 “Lam SH Nat Biotechnol 2007” or p. 19, line 770-772: “… were identify…” should read “…were identified…”).

+ Please remove “Di seguito qulacosa che può essere utile per reference” from p.20, line 803.

Author Response

Comments and Suggestions for Authors

In their review “MicroRNAs in animal models of HCC“, Fornari and colleagues give an extensive overview of the various roles microRNAs (miRNAs) play in animal models of hepatocellular carcinoma (HCC). The authors discuss miRNA knockout and transgenic mouse models, oncogene-related transgenic mice, HCC mouse models suitable for testing miRNA-based therapeutic approaches, xenograft- and patient-derived xenograft mice as well as orthotopic mice. As non-murine models for HCC, several rat models as well as zebrafish models are described. The review is well written and I enjoyed reading this overview. Some comments as mentioned below.

Comments:

+ I would suggest to increase the font size of the mentioned miRNAs in Fig.1, as it was (for me) quite difficult to read, even more so on the print out.

R: As suggested, the font size of Figure 1 has been increased. Moreover, as requested by Reviewer #4, specific references have been added for each miRNA. We thank the Reviewer for this suggestion.

+ I am well aware that a lot of research on HCC and the influence of miRNAs on disease etiology was performed by researchers around Prof. Negrini’s group. However, I would kindly ask to consider a balanced choice of referenced publications. These could include, amongst others, https://doi.org/10.1186/s12885-017-3860-x, https://doi.org/10.1038/s41598-018-30682-5, https://doi.org/10.18632/oncotarget.4202, ...

R: We appreciated the Reviewer’s suggestion and we added the suggested articles. The articles ‘https://doi.org/10.1186/s12885-017-3860-x’ and ‘https://doi.org/10.1038/s41598-018-30682-5’ were added in the ‘HBx transgenic mouse and miR-224’ paragraph. The review article on ‘mouse models of liver cancer’ (https://doi.org/10.18632/oncotarget.4202) was cited in the introductive paragraph to GEM animal models.

+ p3., line 101: I would use “germ line” instead of “germinal” (if I understood the sentence correctly).

R: We thank the Reviewer for this correction. We changed ‘germ line’ with ‘germline’.

+ Although work on HCC in Zebrafish is probably not so extensive, I would try to slightly extend this chapter. As is, this chapter reads a bit as if it was artificially put together from several unrelated paragraphs, and some of them contain errors (e.g. the reference on p. 19 “Lam SH Nat Biotechnol 2007” or p. 19, line 770-772: “… were identify…” should read “…were identified…”).

R: We apologize for errors present in the ‘Zebrafish’ paragraph and we promptly corrected them in the amended version. We extended this paragraph by adding several information about miRNA functions in this model. However, because of Reviewer #4 specific request to eliminate Zebrafish paragraph, we rely upon the Editor’s decision as to whether this paragraph should be delete or not.

+ Please remove “Di seguito qulacosa che può essere utile per reference” from p.20, line 803.

R: We apologize for the Italian sentence that was not cancelled from the manuscript and for the last table that had to be removed from the manuscript, too. We removed both of them in the amended version.

Reviewer 2 Report

Apart from several areas where the English could be improved, this is a very comprehensive overview of the role of microRNAs in HCC development.

Author Response

REviewer 2

Comments and Suggestions for Authors

Apart from several areas where the English could be improved, this is a very comprehensive overview of the role of microRNAs in HCC development.

R: We thank the Reviewer for his/her appreciation of our manuscript. As requested by other Reviewers, some paragraphs, including rat models and concluding remarks, have been changed and a revision of the English language has been performed.

Reviewer 3 Report

It is an extensive review that gathers all the information about existing animal models for the study of miRNA in HCC. The manuscript is very well written and well understood. However, some minor issues should be corrected.
1) There are words in the text with a different format (font and size).
2) Line 579, the Latin names of the plants have to be in italics.
3) In figure 2 it is said that the HCC in humans and rats share common histopathological characteristics. It would be good to describe very briefly what these characteristics are shown in the image.
4) Table 1 provides references that could go into the final reference list.
5) At the end of the text, there is a table without a heading and which is not quoted.
6) The revision is very exhaustive. Consider shortening some sections, such as rat models.

Author Response

Reviewer 3

Comments and Suggestions for Authors

It is an extensive review that gathers all the information about existing animal models for the study of miRNA in HCC. The manuscript is very well written and well understood. However, some minor issues should be corrected.

R: We thank the Reviewer for his/her positive comments and appreciation and we performed all the suggested corrections as below detailed.

1) There are words in the text with a different format (font and size).

R: We carefully checked the entire manuscript and made the text uniform in terms of font and size

2) Line 579, the Latin names of the plants have to be in italics.

R: In the original manuscript, we mentioned the name of the compound and not the name of the plants from which it derived. We better clarified that sentence and we put the name of the plants in italics. We thank the Reviewer for this elucidation

3) In figure 2 it is said that the HCC in humans and rats share common histopathological characteristics. It would be good to describe very briefly what these characteristics are shown in the image.

R: A brief histopathological description of characteristics of rat and human HCCs was added to Figure 2.

4) Table 1 provides references that could go into the final reference list.

R: As requested by Reviewers #3 and #4, the references of Table 1 and 2 were added to the final reference list.

5) At the end of the text, there is a table without a heading and which is not quoted.

R: We apologize for the last table that was indeed not to be included in the present manuscript. We removed it from the amended version. We thank the Reviewer for this correction.

6) The revision is very exhaustive. Consider shortening some sections, such as rat models.

R: As requested by both Reviewer #3 and #4, we shorted the sections regarding the DEN and the R-H rat models, eliminating too descriptive sentences.

Reviewer 4 Report

The aim of this review publication was to “describe different rodent models of HCC emphasizing their representativeness with the human pathology and their usefulness as preclinical tools for assessing miRNA-based therapeutic strategies”. The review is very interesting, and based on many citations of the authors' own works (e.g. 3,34,38,41,56,57, etc.), which gives greater credibility for the compared studies. The paper is important from clinical point of view and may carry practical importance. However, in some places the descriptions are too long and/or off topic (e.g. “R-H rat model”, “Zebrafish”). There is too much publicly known information in the subsection “R-H rat model”, and too few descriptions of the importance of miRNA in the HCC. Because zebrafish do not belong to rodents, but to Cypriniformes in taxonomic hierarchy, I suggest to remove all this subchapter (lines 752-780). This research model does not seem to belong to the objectives of the work. In addition, there are minor errors in this section, e.g. one work has no citation number (line 759). The summary and conclusions are too general, there is a lack of assessment which micro RNAs are the most promising from a clinical point of view and why. I suggest a small rewording of this section.

I also recommend Authors important text corrections that will improve the readability and quality of the work:

1.      Please check precise all the abbreviations especially used for the first time in the text and give the explanations (e g. DNMT1 – line 145 vs. line 529, NAFLD – line 161, WT –line 165, BH3 and BMF – line 179, CCl4  -line 235; MYC, RAS, LIN28B, CTNNB1, SMARC etc –line 300-304; MCC (line 319); Raf/MEK/ERK – abrreviations (line 454); AAV-mediated (line 503); ULK1 (line 507), TACE (line 519), NGS (line 554), PARP (line 789); use the capital letters in case of alpha-SMA, CTGF, TGF-beta1 - lines 248-249; please uniform RAS or Ras, etc. (line 301); it may be better to give a list of abbreviations at the end of the review?

Please standardize the spelling of genes and proteins, e.g. TP53 mutations (line 537); Please change in italic these words: “in vivo” and “in vitro”, “ al.” or standarize everywhere (lines 239, 291, 326, 484, 584, 618, 725, 768); line 579 – names of Astragalus etc. should be also in italic;

4.      In Table 1 and 2, I recommend to give the number of references in contrast to PMID, it is not clear; submitted data – I don’t understand? Table descriptions should be uniform. The last Table (page 21) has no number, and explanations, and is not quoted in the main text. 5.      I recommend to standarize the descriptions of all miRNAs changes (up- and down-regulation) in Figure and both tables (or green/red or italic etc.). 6.      Additionally please improve Figure 1 (please remove the top heading Fig. 1 from the picture, and add the literature based on which the Figure was made).7.      Some small error in line 345: several pro and cons (?).

The paper can be accepted for printing after taking into account most of my comments above.

Author Response

Reviewer 4

The aim of this review publication was to “describe different rodent models of HCC emphasizing their representativeness with the human pathology and their usefulness as preclinical tools for assessing miRNA-based therapeutic strategies”. The review is very interesting, and based on many citations of the authors' own works (e.g. 3,34,38,41,56,57, etc.), which gives greater credibility for the compared studies. The paper is important from clinical point of view and may carry practical importance.

R: We thank the Reviewer for his/her appreciation of our manuscript and for his/her critical comments, which have been addressed in point-by-point specific answers.

+ However, in some places the descriptions are too long and/or off topic (e.g. “R-H rat model”, “Zebrafish”). There is too much publicly known information in the subsection “R-H rat model”, and too few descriptions of the importance of miRNA in the HCC.

R: As requested by both Reviewer #3 and #4, we shorted the sections regarding the DEN and the R-H rat models, eliminating too descriptive sentences. Moreover, a further article reporting the role of miR-375/YAP axis in early hepatocarcinogenesis in the R-H model has been added (REF #182), contributing to explain the key role of miRNA in HCC.

+ Because zebrafish do not belong to rodents, but to Cypriniformes in taxonomic hierarchy, I suggest to remove all this subchapter (lines 752-780). This research model does not seem to belong to the objectives of the work. In addition, there are minor errors in this section, e.g. one work has no citation number (line 759).

R: We apologize for errors present in the ‘Zebrafish’ paragraph and we promptly corrected them in the amended version. Because of Reviewer #1 specific request: ‘Although work on HCC in Zebrafish is probably not so extensive, I would try to slightly extend this chapter’ and due to the novelty and scientific potential of this model for liver cancer research, we decided to improve and slightly extend this paragraph. We leave the decision to the Editor as to whether this paragraph should be delete or not.

+ The summary and conclusions are too general, there is a lack of assessment which micro RNAs are the most promising from a clinical point of view and why. I suggest a small rewording of this section.

R: As suggested by the Reviewer, we revised the conclusions and outlined miRNA-based therapeutic options that in our opinion are the most promising. We thank the Reviewer for this suggestion.

I also recommend Authors important text corrections that will improve the readability and quality of the work:

1.      Please check precise all the abbreviations especially used for the first time in the text and give the explanations (e g. DNMT1 – line 145 vs. line 529, NAFLD – line 161, WT –line 165, BH3 and BMF – line 179, CCl4  -line 235; MYC, RAS, LIN28B, CTNNB1, SMARC etc –line 300-304; MCC (line 319); Raf/MEK/ERK – abrreviations (line 454); AAV-mediated (line 503); ULK1 (line 507), TACE (line 519), NGS (line 554), PARP (line 789); use the capital letters in case of alpha-SMA, CTGF, TGF-beta1 - lines 248-249; please uniform RAS or Ras, etc. (line 301); it may be better to give a list of abbreviations at the end of the review?

R: We added explanations for all abbreviations, uniformed all gene name and included an abbreviation list of the most cited genes and words at the end of the review. We thank the Reviewer for this recommendation.

Please standardize the spelling of genes and proteins, e.g. TP53 mutations (line 537); Please change in italic these words: “in vivo” and “in vitro”, “ al.” or standarize everywhere (lines 239, 291, 326, 484, 584, 618, 725, 768); line 579 – names of Astragalus etc. should be also in italic;

R: We standardized the spelling of genes and proteins and put in italics specific words. We thank the Reviewer for this comment.